

# Brief communication: SWM: Stochastic Weather Model for precipitation-related hazard assessments

Melody G. Whitehead[1], Mark S. Bebbington[1]

[1]Volcanic Risk Solutions, School of Agriculture and Environment, Massey University, Palmerston North, 4472, New Zealand

*Correspondence to*: Melody G. Whitehead (m.whitehead@massey.ac.nz)

**Abstract**

Long-term hazard and risk assessments are produced by combining many hazard-model simulations, each using slightly different set of inputs to cover the uncertainty space. While most input parameters for these models are relatively well-constrained, atmospheric parameters remain problematic unless working on very short-time scales (hours to days).

Precipitation is a key trigger for many natural hazards including floods, landslides, and lahars. This work presents a stochastic catchment-scale weather model that takes openly available ERA5-land data, and produces long-term, spatially varying precipitation data that mimics the statistical dimensions of real-data. This allows precipitation to be robustly included in hazard-model simulations.

## 1 Introduction

Natural hazard and risk assessments are probabilistic by necessity. They must incorporate the intrinsic variability of natural systems, and the large number of unknown (but often data constrained) input parameters. To produce such assessments, many model simulations are run by sampling from a distribution for these parameters. The outputs are then combined (often overlain in a spatial context), to calculate hazard likelihoods across an area, and/or to produce risk maps, key for communicating hazards (Thompson et al., 2015; Hyman et al. 2019). The spatial extent of such hazards is key to such assessments, as is a robust

approach to simulation design. Precipitation is causally linked to many natural hazards including floods, landslides, and lahars (Gill and Malamud, 2016). While several stochastic weather models exist in published literature, they either require detailed local rainfall information – which is rare over long timescales (Zhao et al. 2019; Muñoz-Sabater et al., 2021), or they are run for a single spatial reference point – which is insufficient for many hazard models (e.g., Floods: Arnaud et al., 2002; Landslides: Gao et al., 2017). The model provided here uses openly available ERA5-land data (Muñoz-Sabater, 2019) and produces

realistic (statistically similar) precipitation patterns to improve the sampling strategy of atmospheric properties and support robust hazard assessments. This brief correspondence first presents algorithm construction, and then an example application using the Rangitāiki-Tarawera catchment, Bay of Plenty, New Zealand. All code is in R (R Core Team, 2021), and freely available.



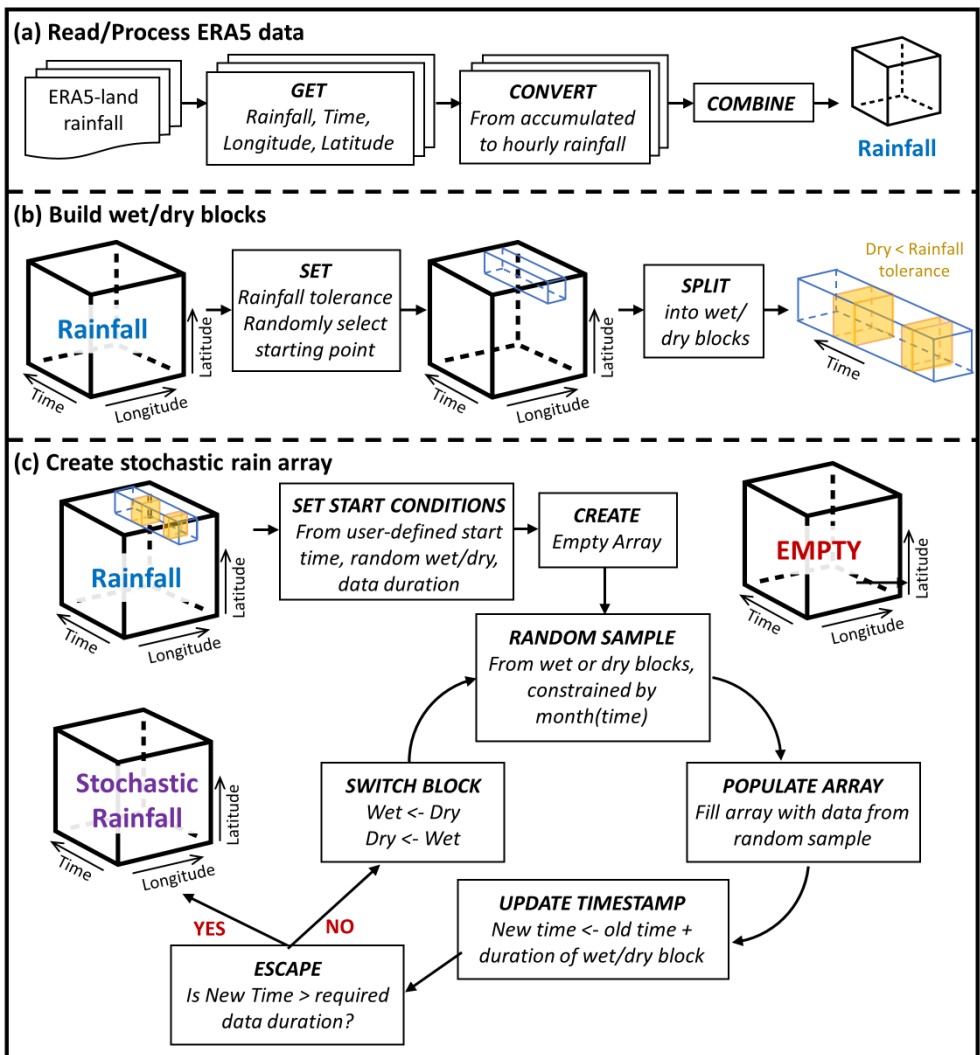

**Figure 1: SWM algorithm flow diagram. (a) Read/Process ERA5 data, (b) Build wet/dry blocks, (c) Create stochastic array.**

## 2 Algorithm construction

The stochastic weather model (SWM) comprises three steps: data conversion, block construction, and stochastic weather generation. Due to the relative simplicity of the model, and exploiting some coding efficiencies in the R package *dplyr* (Wickham et al., 2023), 10 years of hourly data can be generated at points on a 10 by 10 grid on a standard desktop computer

in under five seconds. Before running SWM, data must be downloaded from ERA5-land data in netCDF format. In ERA5-land, the variable is total precipitation *tp* in metres, and is the total amount of water accumulated over a particular time period, resetting every 24 hours (Muñoz Sabater, 2019). SWM will run on as little as one day's worth of rainfall data however, the



outputs will then only mimic the statistical properties of this day's weather. The recommendation is therefore to download at least the last 20 years of data (ERA5 data are available from 1960–current).


SWM first pulls time- and location-stamped precipitation data from the ERA5-land data and converts values from accumulated to hourly rainfall before combining all data into a single spatio-temporal array for analysis (Fig. 1a). A single point is selected at random from the locations used in the download of the original ERA5-land data (Fig. 1b). The precipitation data at this point is used to split the single array into periods of precipitation (*wet*), and no precipitation (*dry*) based on a user-defined rainfall

tolerance (below which an hour is considered *dry*). The user must then define three start conditions: (1) the month and day from which stochastic data are to begin (noting that weather data are seasonal), (2) the length of time data are required for, and (3) how many sets of data are required. For example, 20 datasets for 10 years of data starting from 30th April. SWM builds an empty array, the spatial extent of which is based on the ERA5-land netCDFs, and the user-defined temporal extent. A starting block of wet or dry is randomly selected (constrained by the starting month) and inserted into the empty array, the timestamp

is updated (starting time + block length), a check is made to see whether this timestamp exceeds the required data size, if yes, the algorithm stops (stochastic data already generated), if not, the type of block is switched (wet to dry, or dry to wet), and the loop begins again (Fig. 1c). The final output from SWM is a set of netCDFs identical in form to those of the ERA5-land data except that precipitation is hourly, rather than cumulative.

## 3 Example application: Rangitāiki-Tarawera catchment, Bay of Plenty, New Zealand

The Rangitāiki-Tarawera catchment is an area susceptible to many natural hazards including volcanic eruptions, flooding, and extreme weather events (ex-tropical cyclones). Hourly rainfall data across a 11x14 grid (Fig. 2) of longitude: {176° E, 176.1° E, …, 177.0° E} and latitude: {37.8° S, 37.9° S, …, 39.1° S}, for 40 years (1981 – 2020) were downloaded from ERA5-land. The ERA5-land data is then prepared for the Rangitāiki-Tarawera catchment through SWM by converting these individual netCDFs into a 11 (longitude steps of 0.1° E) by 14 (latitude steps of 0.1° S) by ~350,400 (time steps in hours) array of hourly

data (24 hrs x 365 days x 40 yrs). The Rangitāiki-Tarawera array is then split into wet/dry time-stamped blocks. ERA5-land precipitation data at this catchment were commonly of very small (~$10^{-18}$ m) but non-zero values (common for ERA4-land data, Muñoz-Sabater et al., 2021). The New Zealand climate report for the region (Chapell, 2013) provides average monthly rain and wet days at Kawerau (a town central to the catchment) of 112 wet days per year (~31 %), where wet days are defined as more than one mm per day. A rainfall tolerance level of zero resulted in ~54 % of data classified as *wet*, using the climate

report criteria (less than one mm rainfall in 24 hrs = 4.12 x $10^{-5}$ m hr$^{-1}$) resulted in ~29 % of data defined as *wet*. For this exemplar, the latter was used. Stochastic precipitation data for the Rangitāiki-Tarawera catchment was then built using SWM to obtain 95 sets of 40 years' worth of hourly stochastic rainfall data across the region as 95 netCDFs. The number of runs (95) was chosen here to provide ninety-fifth percentile bounds for the statistical analyses (Sect. 4), but in practice can be set to any value, e.g., to match the number of downstream hazard simulations planned.








**Figure 2: ERA5 data across the case-study area. (a) Case study area, North Island, Aotearoa, basemap from Earthstar Geographics (https://www.terracolor.net/), (b) Elevation of case study area with ERA5 grid, catchment of interest, and Kawerau township, (c) Mean monthly total rainfall for January (ERA5 data: 1981 to 2020), (d) Mean monthly total rainfall for July (ERA5 data: 1981 to 2020).**

## 4 Statistical analyses

Four sets of statistical analyses were undertaken to ensure that SWM simulated data are stochastically similar to ERA5-land data. For this, the 95 sets of 40 years of simulated data across the 11x14 grid are used, code for these analyses being in the GitHub repository, and all outputs for the exemplar in the supplementary material. The location at which tests are performed were selected randomly, and the whole process was run through twice to check sample sensitivity. The latter exercise was to check whether any patterns observed in the first 95 sets of simulated data were repeated in the second set. These are referred to as *Sample 1* and *Sample 2* in the results, with each sample containing 95 sets of simulated data. The analyses were:

    (1) Monthly means and variance

Students t-test was used to compare the monthly mean rainfall between real and simulated data, the Shapiro-Wilks test is used to test normality to determine the appropriate equal variance test: Bartlett if data are normally distributed, or Levene if not (Fox and Weisburg, 2019).

    (2) Significance of month and source for rainfall prediction

Tukey's Honest Significant Difference (HSD) (e.g., Miller 1981) is used to determine whether source (real or simulated) is a significant factor in the prediction of total monthly rainfall.

    (3) Distributions of monthly rainfall totals

A non-parametric bootstrap method is applied whereby empirical cumulative distribution functions (eCDFs) for each simulated dataset representing the 95th percentile envelope (95 runs) is overlain by the real dataset.

    (4) Temporal trends on daily and monthly timescales

Autocorrelations are calculated for each of the real and simulated data to compare any significant lags (Venables and Ripley, 2002). Tests require a continuous variable in discrete time so are run on cumulative daily and cumulative monthly rainfall.

**Figure 3: Statistical analyses results, simulated data are shown as grey lines, real data are shown as red lines.** (a) Sample 1: January empirical Cumulative Distribution Functions of rainfall totals (m), (b) Sample 2: January empirical Cumulative Distribution Functions of rainfall totals (m), (c) Sample 1: July empirical Cumulative Distribution Functions of rainfall totals (m), (d) Sample 2: July empirical Cumulative Distribution Functions of rainfall totals (m), (e) Sample 1: Monthly cumulative rainfall autocorrelations,



**(f) Sample 2: Monthly cumulative rainfall autocorrelations, (g) Sample 1: January daily cumulative rainfall autocorrelations, (h) Sample 2: January daily cumulative rainfall autocorrelations, (i) Sample 1: July daily cumulative rainfall autocorrelations, (j) Sample 2: July daily cumulative rainfall autocorrelations.**

## 5 Results

Overall, while SWM passed all statistical tests for both samples, some departure was noted in several combinations, the specifics of which are detailed below:

    (1)  Monthly means and variance

Real and simulated data failed the Shapiro-Wilks test of normality at a similar rate (~52-58 %) for both samples, thus the Levene test for equal variance was always used.

*Sample 1:* Only eight of 1140 tests showed a statistically significant difference in monthly means ($p < 0.05$); expected number under the null hypothesis of no difference being 57. Fifty-one pairs (5 %) failed the test of equal variance ($p < 0.05$), of which 26 (more than half) were in June.

*Sample 2:* Only three of 1140 tests showed a statistically significant difference in monthly means ($p < 0.05$). Forty-three pairs (4 %) failed the test of equal variance ($p < 0.05$), of which 12 were in June, and 12 in February.

(2)  Significance of month and source for rainfall prediction

Two linear models were built with rainfall as the response variable, and both month and source (real or simulated) as predictor variables. One was built with an interaction term (m1), one without (m2). Both models for both samples passed Tukey's HSD test, rejecting source as a statistically significant predictor (*Sample 1:* $p = 0.8434$; *Sample 2:* $p = 0.9786$).

    (3)  Distribution of monthly rainfall totals

*Samples 1 and 2:* Envelopes were built for each sample and for each month from eCDFs of the simulated data, these were then overlain by the real data, which consistently fell within this ninety-fifth percentile envelope (examples in Fig. 3a-b).

    (4)  Temporal trends on daily and monthly timescales

*Samples 1 and 2:* Autocorrelations for cumulative daily rainfall for both real and simulated data over longer time lags (> 30 days) showed minimal trends through time, thus, although we cannot reject a conclusion of similar trends between the two,

results remain inconclusive. Looking within monthly data only and over shorter time lags (< 30 days), there are potential departures of the real data more positively correlated than the simulated data for January at a lag of 5 days (Fig. 3g-h), and for July at a lag of 8 days (Fig. 3i-j). Autocorrelations at the monthly scale show seasonal trends in both real and simulated data (Fig. 3d) with strong positive correlations at the yearly level (every 12 months), and negative correlations at the six-monthly level (e.g., rainfall in January is negatively correlated with rainfall in July). A potential horizontal offset is noted between real

and simulated data. The differences between real and simulated data here are attributed to multi-day rain or dry events. For example, in the simulated data, a large *wet* block representing a tropical cyclone in the original data occurring entirely in December, can feasibly start at 31 December in the simulated data and thus will run over into January. January and July results are provided here as they represent the most extreme weather months for the region (in terms of both drought and heavy





rainfall) but also the most extreme departures from the simulated data. The remainder of the months do not show any notable

differences in lags between simulated and real data for either sample (see supplementary material for complete result set).

**6 Conclusions**

The method and code provided through this brief communication can be used to generate multiple sets of realistic, long-term, hourly precipitation data over a spatial region. This code provides an easy to plug-in input for hazard simulations to support long-term, time and spatially varying, probabilistic risk assessments, uncertainty quantification, and multi-hazard models.

**Data availability**

All data were obtained from the Copernicus Climate Change Service (CS3) Climate data Store (CDS) and is published under a Creative Commons Attribution 4.0 International (CC BY 4.0) license.

**Code availability**

Code is written in R (open-source software) and is freely available at https://github.com/MelWhitehead/SWM.

**Supplement**

The supplement related to this article is available online at: << link for nhess supplement to go here >>

**Author contributions**

Both authors conceptualised the model, MW built the model, statistical tests were guided by MB and coded by MW.

**Competing interests**

The authors declare that they have no conflict of interest.

**Acknowledgements**

This research was supported by Resilience to Nature's Challenges Multi-hazard Risk model Program, New Zealand (contract GNS-RNC043).



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
