# Peer review of "Brief communication: SWM: Stochastic Weather Model for precipitation-related hazard assessments using ERA5-land data"

_Natural Hazards and Earth System Sciences, 2023_

## Referee Comment (RC2)

This manuscript proposes a catchment-scale stochastic weather model to generate precipitation ensembles using hourly ERA5-Land precipitation data as input. While the authors state that the model is validated for the Rangitāiki-Tarawera catchment, New Zealand, it seems that the results are actually limited to only a few grid boxes. Several critical issues regarding stochastic modeling of precipitation and its validation remain unaddressed. I detail the major concerns below for the authors' consideration.

Major comments:

1. Methodology (Section 2 and Fig. 1): the generation of stochastic rainfall series only involves the selection of a location or point (longitude, latitude), and the random resampling and combination of the identified wet and dry blocks along the time dimension. For this procedure, two key issues need clarification in terms of space-time statistics: (a) for the point-scale temporal process, how can this resampling and combination operation maintain long-term climatology (from ERA5-Land)? e.g., the point-scale climatological mean, rainy days/hours, and other moments that characterize the time process; (b) from the space process perspective, how does this model account for the spatial correlation between the generated point-scale time series? This is particularly important for catchment-scale applications. The authors might want to check out these reference papers for a better understanding of the two aspects: Waymire, E. and Gupta, V. K., WRR, 1987; A. Burton et al., Environ. Model. Softw., 2008; D. Kim and C. Onof., JoH, 2020.

2. Statistical validation (Sections 4 & 5): it's unclear how many grid boxes are used in the validation. For example, in Fig. 3 the authors should indicate whether the results are for a randomly selected grid, or average across all grids (11*14 grids). Additionally, the authors should include figures to verify the spatial pattern derived from the model: e.g., to compare maps of the model-derived 40-year mean (75%, 95% quantiles, and rainy days/hours) with the ERA5-Land reference statistics. Lastly, the absence of a split of training and validation datasets needs justified.

3. The Introduction section highlights that this model can "produce realistic precipitation patterns to improve the sampling of atmospheric properties and support robust hazard assessments". I'd argue that for most hazard assessments, a model for real-time precipitation ensembles is typically required (see Samantha H. Hartke. WRR, 2022 for a review). Can this model serve that purpose? If not, the authors should clearly explain how information from this stochastic rainfall model can be ingested into hazard assessments.

Minor comments:

4. Abstract: please revise this part, to include the contribution and fundamentals of the stochastic model, the study period (1981-2020 ERA5-Land data), the validation catchment, and how the modeled information can be used in precipitation-related hazard assessments.

5. Figure 1: the rectangle "COMBINE" is not clear – it would be better if described as "combination of time, longitude and latitude dimensions to build a 3-D rainfall cube". In addition, please clarify the term "starting point" in Figure 1(b).

6. Lines 37-39: I cannot quite follow this sentence. Does this model only generate rainfall statistics for a specific day? And "ERA5 data" should be "ERA5-Land".

7. Lines 46-47: I don't understand why the model needs to specify the start date as 30th April. For example, what's the difference between the 10-year data starting from 30th April and 10-year data starting on 31st July.

8. Lines 67-69: What's the rationale behind conducting 95 runs for 95th percentile bounds? I mean, the 95th percentile can be also estimated from 100, 1000, or 5000 runs. In those cases, the model-derived 95th percentile might be more statistically robust due to the increased number of runs.

9. Section 4: this section describes how to validate the model; a more appropriate title might be "Evaluation Method" or "Validation Method". In addition, if "Sample 1" and "Sample 2" refer to two independent runs of the model at a specific gridbox, it would be better to use "Realization 1" and "Realization 2".

10. Conclusions: I don't think the presented work sufficiently supports these conclusions, as many points lack supporting results. For instance, the monthly and daily validation results (despite their flaws as noted above) cannot support the claim that the model can "generate realistic, long-term, hourly precipitation data".

References:

[1] Burton, A., Kilsby, C. G., Fowler, H. J., Cowpertwait, P. S. P., & O'Connell, P. E. (2008). RainSim: A spatial–temporal stochastic rainfall modelling system. Environmental Modelling & Software, 23(12), 1356–1369.

[2] Hartke, S. H., Wright, D. B., Li, Z., Maggioni, V., Kirschbaum, D. B., & Khan, S. (2022). Ensemble Representation of Satellite Precipitation Uncertainty Using a Nonstationary, Anisotropic Autocorrelation Model. Water Resources Research, 58(8).

[3] Kim, D., & Onof, C. (2020). A stochastic rainfall model that can reproduce important rainfall properties across the timescales from several minutes to a decade. Journal of Hydrology, 589.

[4] Waymire, E., & Gupta, V. K. (1981). The mathematical structure of rainfall representations: 1. A review of the stochastic rainfall models. Water Resources Research, 17(5), 1261–1272.

---

## Author Comment (AC2)

**Comments from Referee 2:**
**Citation**: https://doi.org/10.5194/nhess-2023-160-RC2

This manuscript proposes a catchment-scale stochastic weather model to generate precipitation ensembles using hourly ERA5-Land precipitation data as input. While the authors state that the model is validated for the Rangitāiki-Tarawera catchment, New Zealand, it seems that the results are actually limited to only a few grid boxes. Several critical issues regarding stochastic modelling of precipitation and its validation remain unaddressed. I detail the major concerns below for the authors' consideration.

Major comments:

1. Methodology (Section 2 and Fig. 1): the generation of stochastic rainfall series only involves the selection of a location or point (longitude, latitude), and the random resampling and combination of the identified wet and dry blocks along the time dimension. For this procedure, two key issues need clarification in terms of space-time statistics:
(a) for the point-scale temporal process, how can this resampling and combination operation maintain long-term climatology (from ERA5-Land)? e.g., the point-scale climatological mean, rainy days/hours, and other moments that characterize the time process;

We can assemble supplementary material based on Figures 2b, 7, and 9 in Kim and Onof (2020) suggested by the Reviewer that will address this point.

(b) from the space process perspective, how does this model account for the spatial correlation between the generated point-scale time series? This is particularly important for catchment-scale applications. The authors might want to check out these reference papers for a better understanding of the two aspects: Waymire, E. and Gupta, V. K., WRR, 1987; A. Burton et al., Environ. Model. Softw., 2008; D. Kim and C. Onof., JoH, 2020.

The method is proposed for use in multihazard cascade modelling at short spatial scales [needs to be emphasised in revision]. The key issue is the ability to rapidly and efficiently generate vast numbers of longitudinal series, using a complexity fit for purpose. Hence spatial variation is handled in a simplistic fashion whereby one location controls the temporal pattern of synthetic data, and data at all other locations are transplanted with it via means of the original timestamp. In solely hydrologically focussed settings, a more complex model such as Burton et al. (2008), or Papalexiou (2022) would be a more appropriate choice. This would be discussed in the revision, subject to space considerations as the Editor might permit.

2. Statistical validation (Sections 4 & 5): it's unclear how many grid boxes are used in the validation. For example, in Fig. 3 the authors should indicate whether the results are for a randomly selected grid, or average across all grids (11*14 grids). Additionally, the authors should include figures to verify the spatial pattern derived from the model: e.g., to compare maps of the model-derived 40-year mean (75%, 95% quantiles, and rainy days/hours) with

the ERA5-Land reference statistics. Lastly, the absence of a split of training and validation datasets needs justified.

The results in Figure 3 are shown for a single randomly selected grid point, the checks, however, were done for all locations. We agree this should be indicated more clearly in the text.
We are happy to include the suggested figures on spatial pattern if the reviewer still thinks this is necessary after our response to 1(b) above. However, these would need to go into supplementary material due to figure number limitations.
To accomplish the objective in 1(b) above, long-term climate trends are not included in the model, hence a training / validation split was not considered. The methodology can be executed on any temporally contiguous subset of ERA5-land to reproduce the desired level of rainfall variation.

3. The Introduction section highlights that this model can "produce realistic precipitation patterns to improve the sampling of atmospheric properties and support robust hazard assessments". I'd argue that for most hazard assessments, a model for real-time precipitation ensembles is typically required (see Samantha H. Hartke. WRR, 2022 for a review). Can this model serve that purpose? If not, the authors should clearly explain how information from this stochastic rainfall model can be ingested into hazard assessments.

The hazard space that we work within is predominantly multi-hazard, including volcanology, seismology, flooding, and geomorphology to name a few. For these, we are looking at long-term probabilities and consequent risks/impacts. Thus, our aim is to produce realistic (but not real / forecasted) rainfall data over a region of interest over long periods of time very quickly so that we can run our downstream models (e.g., tephra re-dispersal, lahar triggering and characteristics etc.) many times with varying but plausible rainfall datasets. This model is currently used within a multi-hazard assessment for the catchment highlighted in this manuscript, but this multi-hazard assessment is still in progress and thus not yet published in any peer-reviewed journal (https://resiliencechallenge.nz/multihazard-risk-model-flooding-case-study-simulation/).

Minor comments:
4. Abstract: please revise this part, to include the contribution and fundamentals of the stochastic model, the study period (1981-2020 ERA5-Land data), the validation catchment, and how the modeled information can be used in precipitation-related hazard assessments.

No problem.

5. Figure 1: the rectangle "COMBINE" is not clear – it would be better if described as "combination of time, longitude and latitude dimensions to build a 3-D rainfall cube". In addition, please clarify the term "starting point" in Figure 1(b).

Happy to replace "COMBINE: with the reviewers extended description.

"Starting point" is the random selection of the point within the 11 x 14 grid that drives the model (i.e., the one that is used to split the time series into wet and dry blocks). Perhaps a different term would be clearer – "driver location" perhaps?

6. Lines 37-39: I cannot quite follow this sentence. Does this model only generate rainfall statistics for a specific day? And "ERA5 data" should be "ERA5-Land".

Sentence removed as reviewer is correct, it is confusing and unnecessary. ERA5 data changed to ERA5-Land data.

7. Lines 46-47: I don't understand why the model needs to specify the start date as 30th April. For example, what's the difference between the 10-year data starting from 30th April and 10-year data starting on 31st July.

This is a downstream user detail – as noted elsewhere, the aim for this rainfall model is to support long-term probabilistic hazard analyses, if an eruption occurs during a dry season, it is less likely to trigger lahars (volcanic mudflows), than if it occurs during a wet season. A start-date also provides opportunities to test reproducibility of downstream-hazard model results.

8. Lines 67-69: What's the rationale behind conducting 95 runs for 95th percen2le bounds? I mean, the 95th percentile can be also estimated from 100, 1000, or 5000 runs. In those cases, the model derived 95th percentile might be more statistically robust due to the increased number of runs.

Subject to both reviewers' responses, we are happy to run, e.g., 999 runs and pull the 95$^{th}$ percentiles from this. The reason we did not do this for initial submission was because the memory allocation of 95 * 40 years of hourly data at 11 x 14 locations was close to the operating capacity of the then-standard desktop computer used by Whitehead, so we thought this would be easier to replicate.

9. Section 4: this section describes how to validate the model; a more appropriate title might be "Evaluation Method" or "Validation Method". In addition, if "Sample 1" and "Sample 2" refer to two independent runs of the model at a specific gridbox, it would be better to use "Realization 1" and "Realization 2".

Agree – happy to change section title to "Evaluation Method", and "Sample" to "Realisation."

10. Conclusions: I don't think the presented work sufficiently supports these conclusions, as many points lack supporting results. For instance, the monthly and daily validation results (despite their flaws as noted above) cannot support the claim that the model can "generate realistic, long-term, hourly precipitation data".

The original conclusions were written in favour of brevity and thus didn't include a lot of the qualifiers or context that otherwise may have been included.

What about something like the following? Added or altered text is underlined.

"The method and code provided through this brief communication can be used to rapidly generate multiple sets of realistic, long-term, hourly precipitation data over a spatial region. While the outputs do not have the nuances that come with more complex models, the efficient open-source code, written in an open-source language, and based on open-source data facilitate an easy to plug-in input for hazard simulations to support long-term, time and spatially varying, probabilistic risk assessments, uncertainty quantification, and multi-hazard models."

---

## Author Response (AR1)

**Response to Reviewers:**

Thanks for taking the time to properly go over our original manuscript. We have now greatly expanded the supplementary material to include multiple different analyses of our simulated results as suggested by Referee 2 (with great tangible examples, thanks!). We have also re-run everything with 999-runs per realisation, which took us a bit longer than expected, so your patience has also been greatly appreciated while we ensured we had run all requested tests and produced the relevant figures.

Mel & Mark.

**Comments from Referee 1:**
**Citation**: https://doi.org/10.5194/nhess-2023-160-RC1

In this MS, the authors discussed the capabilities of stochastic weather models on predicting rainfall in the Rangitāiki-Tarawera catchment. They have demonstrated the potential of SWM based on the ERA5-land data. However, some issues need to be addressed:

The author is supposed to add the "ERA5-land data" in the title, on which this MS is based.
New proposed title: "Brief communication: SWM: Stochastic Weather Model for precipitation-related hazard assessments using ERA5-land data"

In Lines 41-42, I don't get the point "converts values from accumulated to hourly rainfall", free hourly precipitation data can be downloaded from the ERA5 website.
ERA5-land data provides precipitation data at the hourly level *but* this is accumulated over 24 hours ending at 00 UTC, this is different to the single-level data ERA5 which is potentially causing the confusion. See for example the documentation on Accumulations: https://confluence.ecmwf.int/display/CKB/ERA5-Land%3A+data+documentation#ERA5Land:datadocumentation-accumulationsAccumulations

In Lines 68, 95 sets are obtained to provide ninety-fifth percentile bounds. According to the MS, the more sets the better results. The authors need to explain why they had to generate 95 sets?
The more sets ran, the more confident we can be in the results, because we have a better idea of the total answer space in which the simulated data can lie. We suggest the addition of information about standard practice for bootstrap significance testing as: "*this is common practice to assess statistical significance with non-parametric bootstrap methods (DiCiccio & Efron, 1996; Ramachandran & Tsokos, 2021).*"

Additionally, we have now run 999 simulations and pulled the 95th percentiles from this. The reason we did not do this for initial submission was because the memory allocation of 95 * 40 years of hourly data at 11 x 14 locations was close to the operating capacity of the then-standard desktop computer used by Whitehead, so we thought this would be easier to replicate by readers / reviewers.

Fig. 3, the authors are suggested to draw the ninety-fifth percentile bounds at (e) to (j).
Figure 3 caption text has been updated to include: "Ninety-fifth percentile bounds for (e) to (j) are represented by the envelope built from the simulated data (grey lines)"
We note that as each set of simulated data include some variation, every time this exercise is run we would see slightly different results which is why we ran two sets (sample 1 and sample 2), with extra results and variations between the two shown in the supplementary material.

The above details have also changed after performing the 999-run update.

Fig. 3, The ACF values approximate 0 over time, maybe adding a table could better illustrate the results.
The figure/table limit is already reached for Brief Communications for NHESS, and we feel that temporal trends are better visualised as plots against time. However, we can consider adding this to supplementary material if the reviewer feels strongly about this, but would request more clarity about how the reviewer feels this could best be presented as a table.

Fig. 3, some small mistakes in the Y-AXIS of (f) and (h).
The whole figure was redone with 999 runs (but we appreciate your attention to detail).

**Comments from Referee 2:**
**Citation**: https://doi.org/10.5194/nhess-2023-160-RC2

This manuscript proposes a catchment-scale stochastic weather model to generate precipitation ensembles using hourly ERA5-Land precipitation data as input. While the authors state that the model is validated for the Rangitāiki-Tarawera catchment, New Zealand, it seems that the results are actually limited to only a few grid boxes. Several critical issues regarding stochastic modelling of precipitation and its validation remain unaddressed. I detail the major concerns below for the authors' consideration.

Major comments:

1. Methodology (Section 2 and Fig. 1): the generation of stochastic rainfall series only involves the selection of a location or point (longitude, latitude), and the random resampling and combination of the identified wet and dry blocks along the time dimension. For this procedure, two key issues need clarification in terms of space-time statistics:
(a) for the point-scale temporal process, how can this resampling and combination operation maintain long-term climatology (from ERA5-Land)?  e.g., the point-scale climatological mean, rainy days/hours, and other moments that characterize the time process;
Supplementary material has been assembled based on Figures 2b, 7, and 9 in Kim and Onof (2020) suggested by the Reviewer that addresses this point.

(b) from the space process perspective, how does this model account for the spatial correlation between the generated point-scale time series? This is particularly important

for catchment-scale applications. The authors might want to check out these reference papers for a better understanding of the two aspects: Waymire, E. and Gupta, V. K., WRR, 1987; A. Burton et al., Environ. Model. Softw., 2008; D. Kim and C. Onof., JoH, 2020.

The method is proposed for use in multihazard cascade modelling at short spatial scales [this is now also emphasised in the revised text]. The key issue is the ability to rapidly and efficiently generate vast numbers of longitudinal series, using a complexity fit for purpose. Hence spatial variation is handled in a simplistic fashion whereby one location controls the temporal pattern of synthetic data, and data at all other locations are transplanted with it via means of the original timestamp. In solely hydrologically focussed settings, a more complex model such as Burton et al. (2008), or Papalexiou (2022) would be a more appropriate choice. This is now included in the conclusion text, alongside references to these models.

2. Statistical validation (Sections 4 & 5): it's unclear how many grid boxes are used in the validation. For example, in Fig. 3 the authors should indicate whether the results are for a randomly selected grid, or average across all grids (11*14 grids). Additionally, the authors should include figures to verify the spatial pattern derived from the model: e.g., to compare maps of the model-derived 40-year mean (75%, 95% quantiles, and rainy days/hours) with the ERA5-Land reference statistics. Lastly, the absence of a split of training and validation datasets needs justified.

The results in Figure 3 are shown for a single randomly selected grid point, the checks, however, were done for all locations. We agree this should have been indicated more clearly in the text and is now clarified in the Figure 3 caption, as well as location information given through the addition of white and pink boxes to Figure 2. We feel that our response to 1(b) above alongside the (many!) additional figures in the supplementary material sufficiently cover this point. To accomplish the objective in 1(b) above, long-term climate trends are not included in the model, hence a training / validation split was not considered. The methodology can be executed on any temporally contiguous subset of ERA5-land to reproduce the desired level of rainfall variation.

3. The Introduction section highlights that this model can "produce realistic precipitation patterns to improve the sampling of atmospheric properties and support robust hazard assessments". I'd argue that for most hazard assessments, a model for real-time precipitation ensembles is typically required (see Samantha H. Hartke. WRR, 2022 for a review). Can this model serve that purpose? If not, the authors should clearly explain how information from this stochastic rainfall model can be ingested into hazard assessments. The hazard space that we work within is predominantly multi-hazard, including volcanology, seismology, flooding, and geomorphology to name a few. For these, we are looking at long-term probabilities and consequent risks/impacts. Thus, our aim is to produce realistic (but not real / forecasted) rainfall data over a region of interest over long periods of time very quickly so that we can run our downstream models (e.g., tephra re-dispersal, lahar triggering and characteristics etc.) many times with varying but plausible rainfall datasets. This model is currently used within a multi-hazard assessment for the catchment highlighted in this manuscript, but this multi-hazard assessment is still in progress and thus not yet published in any peer-reviewed journal (https://resiliencechallenge.nz/multihazard-risk-model-flooding-case-study-simulation/).

Minor comments:

4. Abstract: please revise this part, to include the contribution and fundamentals of the stochastic model, the study period (1981-2020 ERA5-Land data), the validation catchment, and how the modeled information can be used in precipitation-related hazard assessments.
Done.

5. Figure 1: the rectangle "COMBINE" is not clear – it would be better if described as "combination of time, longitude and latitude dimensions to build a 3-D rainfall cube". In addition, please clarify the term "starting point" in Figure 1(b).
COMBINE replaced in figure with "BUILD: *Combine Time, Longitude, Latitude.*" "3-D" also added to the in-text description.
"Starting point" is the random selection of the point within the 11 x 14 grid that drives the model (i.e., the one that is used to split the time series into wet and dry blocks). We have replaced "starting point" with "driver location" in the figure.

6. Lines 37-39: I cannot quite follow this sentence. Does this model only generate rainfall statistics for a specific day? And "ERA5 data" should be "ERA5-Land".
Sentence removed as reviewer is correct, it is confusing and unnecessary. ERA5 data corrected to ERA5-Land data throughout.

7. Lines 46-47: I don't understand why the model needs to specify the start date as 30th April. For example, what's the difference between the 10-year data starting from 30th April and 10-year data starting on 31st July.
This is a downstream user detail – as noted elsewhere, the aim for this rainfall model is to support long-term probabilistic hazard analyses, if an eruption occurs during a dry season, it is less likely to trigger lahars (volcanic mudflows), than if it occurs during a wet season. A start-date also provides opportunities to test reproducibility of downstream-hazard model results.

8. Lines 67-69: What's the rationale behind conducting 95 runs for 95th percen2le bounds? I mean, the 95th percentile can be also estimated from 100, 1000, or 5000 runs. In those cases, the model derived 95th percentile might be more statistically robust due to the increased number of runs.
As suggested, we have now produced results for two realisations, each of 999 runs of 40 years' worth of yearly data and pulled the 95th percentiles from this. The reason we did not do this for initial submission was because the memory allocation of 95 * 40 years of hourly data at 11 x 14 locations was close to the operating capacity of the then-standard desktop computer used by Whitehead, so we thought this would be easier to replicate.

9. Section 4: this section describes how to validate the model; a more appropriate title might be "Evaluation Method" or "Validation Method". In addition, if "Sample 1" and "Sample 2" refer to two independent runs of the model at a specific gridbox, it would be better to use "Realization 1" and "Realization 2".
Agreed. Section title changed to "Evaluation Method", and all "Samples" to "Realisations".

10. Conclusions: I don't think the presented work sufficiently supports these conclusions, as many points lack supporting results. For instance, the monthly and daily validation results (despite their flaws as noted above) cannot support the claim that the model can "generate realistic, long-term, hourly precipitation data".

The original conclusions were written in favour of brevity and thus didn't include a lot of the qualifiers or context that otherwise may have been included. We have rewritten these conclusions as follows (underlined text is new):
"The method and code provided through this brief communication can be used to rapidly generate multiple sets of realistic, long-term, hourly precipitation data over a spatial region. While the outputs do not have the nuances that come with more complex models, the efficient open-source code, written in an open-source language, and based on open-source data facilitate an easy to plug-in input for hazard simulations to support long-term, time and spatially varying, probabilistic risk assessments, uncertainty quantification, and multi-hazard models."

---

## Author Response (AR2)

**Response to Reviewers:**

Thanks for taking the time to (re)review our manuscript, we believe everything requested has been addressed.

Mel & Mark.

**Reviewer 1:**

The writing should be further improved and the data should be double checked.
Data have been double checked via a second download from ERA5-land and are correct. Please see tracked-changes for specific text-related improvements.

**Reviewer 2:**

I appreciate the efforts the authors have made to address all my concerns. The manuscript now seems clearer and more reasonable to me. After addressing a few minor issues as listed below, this paper can be accepted for publication.

(1) Since the spatial correlation issue has not been addressed in the SMW model, I suggest removing the term "watershed-scale weather model" from the Abstract and elsewhere in the paper.
Done.

(2) caption of Fig.1: Change "Read/Process ERA5 data" to "Read/Process ERA5-land data".
Done.

(3) Please add at least one sentence to introduce the input ERA5-land data, including its spatial resolution (0.1°), time span, and the official download link.
Done. Sentence included at lines 39/40 and reads: *"Spatial resolution is 0.1°, and data are available from January 1950, at https://cds.climate.copernicus.eu/cdsapp#!/dataset/reanalysis-era5-land "*

(4) Lines 49-51 (in the changes-tracked version): Since the authors mentioned the example of "20 datasets for 10 years of data starting from April 30th," should the following sentence be: "...the starting block of wet or dry is randomly selected (constrained by the starting date)" as April 30th, rather than "starting month"? Then, what is the "starting date" for your working example in Section 3? Please explain in Section 3 and add this detail.
Thanks, these clarifications have been added in - see lines 48 to 50 for method description update, and 68 for example application clarification.

(5) Given that the wet-dry threshold is nonzero (less than one mm of rainfall in 24 hours = $4.12 \times 10^{-5}$ m/hr; in Section 3), I think the wet threshold used for calculating the portion of wet periods should also not be 0 m? (the lower-right panel of Supplementary Figure 5).
Agreed, this was how the plots were produced, but as there was no discernible difference in results between > 0 m and > (1/24)/1000 m, we used > 0m as it looked cleaner. Figure captions now altered to include this detail.